# Multiple Beneficial Effects of Aloesone from *Aloe vera* on LPS-Induced RAW264.7 Cells, Including the Inhibition of Oxidative Stress, Inflammation, M1 Polarization, and Apoptosis

**DOI:** 10.3390/molecules28041617

**Published:** 2023-02-08

**Authors:** Yan Wang, Zhongyv Xiong, Chang Li, Dong Liu, Xiaogang Li, Junyv Xu, Niangen Chen, Xuesong Wang, Qifu Li, Youbin Li

**Affiliations:** 1Key Laboratory of Tropical Translational Medicine of Ministry of Education, Hainan Key Laboratory for Research and Development of Tropical Herbs, Key Laboratory of Li Nationality Medicine, School of Pharmacy, Department of Neurology, The First Affiliated Hospital of Hainan Medical University, Hainan Medical University, Haikou 571199, China; 2Key Laboratory of Brain Science Research & Transformation in Tropical Environment of Hainan Province, Department of Neurology, The First Affiliated Hospital of Hainan Medical University, Hainan Medical University, Haikou 571100, China

**Keywords:** aloesone, anti-apoptosis, anti-inflammation, anti-M1 polarization, antioxidant stressK

## Abstract

Aloesone is a major metabolic compound in *Aloe vera*, which has been widely used as a food source and therapeutic agent in several countries. Our recent study demonstrated that aloesone has anti-epileptic effects on glutamate-induced neuronal injury by suppressing the production of reactive oxygen species (ROS). Unless ROS are naturally neutralized by the endogenous antioxidant system, they lead to the activation of inflammation, polarization, and apoptosis. This study aimed to identify the multiple beneficial effects of aloesone and explore its molecular mechanism in macrophages. Hence, the murine macrophage cell line RAW264.7 was pretreated with aloesone and then exposed to lipopolysaccharides (LPS). The results demonstrated that aloesone, within a dosage range of 0.1–100 µM, dramatically decreased the LPS-induced elevation of ROS production, reduced nitric oxide (NO) release, inhibited the M1 polarization of RAW264.7 cells, and prevented cells from entering the LPS-induced early and late phases of apoptosis in a dose-dependent manner. Simultaneously, aloesone significantly decreased the mRNA expression of inflammation-related genes (iNOS, IL-1ꞵ, TNF-α) and increased the expression of antioxidant enzymes (Gpx-1 and SOD-1). The core genes HSP90AA1, Stat3, Mapk1, mTOR, Fyn, Ptk2b, and Lck were closely related to these beneficial effects of aloesone. Furthermore, immunofluorescence staining and flow cytometry data confirmed that aloesone significantly repressed the activation of mTOR, p-mTOR, and HIF-1α induced by LPS and inhibited the protein expression of TLR4, which is the target of LPS. In conclusion, aloesone demonstrated multiple protective effects against LPS-induced oxidative stress, inflammation, M1 polarization, and apoptosis in macrophages, suggesting its potential as a prodrug.

## 1. Introduction

Inflammation is the host’s immune response to chemicals, physical injury, and infection. However, excessive inflammation induces an unpredicted increase in inflammatory mediators, such as cytokines (tumor necrosis factor-α (TNF-α), interleukins, inducible nitric oxide synthase (iNOS)), chemokines, and reactive oxygen species (ROS). Inflammatory mediators have implications in heart disease (ischemic heart failure and cardiac ischemia/reperfusion injury) [1,2], brain disorders (depression and anxiety) [3,4], and lung disease (chronic obstructive pulmonary disease and acute lung injury) [5,6]. Recent studies have confirmed that ROS could also cause oxidative stress, leading to the activation of inflammatory pathways, the stimulation of macrophage polarization, and the triggering of cell damage [7,8].

Traditionally, *Aloe vera* has been widely used as a food source and therapeutic agent in several countries, such as Egypt, India, Greece, and China [9,10]. Various compounds from A. vera, including anthrones, alkaloids, anthraquinones, chromones, and flavonoids [11], have been shown to exhibit anti-tyrosinase [12], anti-cancer [13,14], anti-diabetic [15], and anti-inflammation effects. Overall, the ethanol extraction of A. vera leaves inhibited the release of interleukin (IL)-6 induced by the neuropeptide substance in glioblastoma/astrocytoma U373 MG cells [16] and significantly suppressed inflammatory cell infiltration and serum secretion of TNF-α and nitric oxide (NO) in rats with trinitrobenzenesulfonic acid-induced colitis [17]. Additionally, gavaging A. vera gel for eight weeks could inhibit hepatic malondialdehyde and lower glutathione levels in high fat, high fructose-diet fed rats [18]. Notably, 100–200 µg/mL of aloin dramatically decreased the release of cytokines and ROS in lipopolysaccharide (LPS)-induced macrophages [19]. Concentrations of aloe emodin from 25 to 100 µM inhibited palmitic acid-induced inflammatory cytokine (TNF-α, IL-1ꞵ, and IL-6) production in H9C2 cells [20]. Furthermore, 80 and 150 mg/Kg of aloe emodin suppressed sepsis-associated inflammatory cytokine production, including TNF-α and IL-6 [21]. Barbaloin, the major anthraquinone in A. vera, ameliorated dextran sulfate sodium salt (DSS)-induced excessive release of TNF-α, IL-1ꞵ, IL-6, and IL-4 [22]. Aloesone, a major metabolic compound in A. vera, shows both antioxidation and inflammation in vitro [23]. Our recent study demonstrated that aloesone caused an anti-epileptic effect in glutamate-induced neuron injury by suppressing the production of ROS [24]. Unless ROS are naturally neutralized by the endogenous antioxidant system, they lead to the activation of inflammation. Hence, we hypothesized that aloesone can potentially exert antioxidant and anti-inflammation effects in peripheral tissues. 

Macrophages are a central component of the innate peripheral immune system and play a vital role in inflammation [25]. The membranes of Gram-negative bacteria contain LPS, which activates the host Toll-like receptor 4 (TLR4) and triggers an inflammatory response, ultimately leading to the release of pro-inflammatory mediators [26,27]. The LPS-induced murine macrophage cell line RAW264.7 model has been commonly used to explore inflammation, oxidative stress, and apoptosis [28,29,30]. The aim of the present study was to explore the effects of aloesone on LPS-induced macrophages by evaluating the main markers of oxidative stress, inflammation, polarization, and apoptosis. The molecular mechanism of aloesone was further studied.

## 2. Results

### 2.1. Aloesone Inhibited LPS-Induced Oxidative Stress in RAW264.7 Cells

Aloesone concentrations of 0.1, 1, 10, 100, and 1000 µM were used to verify its effect on the survival of RAW264.7 cells using the Cell Counting Kit-8 (CCK8). Results demonstrated that these concentrations of aloesone did not affect the survival of RAW264.7 cells (Figure 1A,B). According to our previous study, concentrations varying from 0.1 to 100 µM of aloesone were shown to ameliorate glutamate-induced neuron injury, and these concentrations were applied in subsequent experiments [24]. 

In the present study, pretreatment with aloesone for 2 h dramatically reduced the LPS-stimulated elevation of ROS production in a dose-dependent manner (Figure 1C,D). In contrast, aloesone significantly increased the mRNA expression of Gpx-1 (Figure 1E) and SOD-1 (Figure 1F), which could clear the overloaded ROS, compared with the LPS group. These results confirmed the antioxidant effect of aloesone.

### 2.2. Aloesone Suppressed Inflammation Induced by LPS

In the present study, the NO level was significantly increased in LPS-induced RAW264.7 cells (11.62 ± 0.38 µg/mL), compared with that in the control group (4.49 ± 0.33 µg/mL). Aloesone decreased the NO release induced by LPS (the 0.1, 1, 10, and 100 µM aloesone doses corresponded to 10.94 ± 0.37, 11.17 ± 0.48, 10.82 ± 0.50, and 8.90 ± 0.48 µg/mL NO, respectively, Figure 2A) and suppressed the mRNA expression of inflammatory cytokines, including iNOS (Figure 2B), IL-1β (Figure 2C), and TNF-α (Figure 2D). These results suggested that aloesone caused anti-inflammatory effects in the RAW264.7 macrophage.

### 2.3. Aloesone Inhibited the M1-Polarization of RAW 264.7 Cells Induced by LPS

As shown in the micrographs (Figure 3A), the administration of LPS for 12 h stimulated the polarization of RAW264.7 cells, with apparent antenna, a characteristic of the M1 phenotype, while aloesone inhibited this polarization. Furthermore, we confirmed the effect of aloesone on the polarization of RAW264.7 cells by detecting the specific surface phenotype marker of M1 (cluster of differentiation, CD86) [31]. The results demonstrated that LPS induced the membrane overexpression of CD86, while aloesone significantly inhibited the membrane expression of CD86, indicating that aloesone inhibited the polarization of RAW264.7 to M1 (Figure 3B). 

### 2.4. Aloesone Suppressed LPS-Induced Apoptosis in RAW 264.7 Cells

In the present study, the administration of aloesone within the 0.1 to 100 µM dosage range prevented the LPS-induced early phase of apoptosis (4.42 ± 0.70%) in a dose-dependent manner (0.1 µM, 3.58 ± 1.15%; 1 µM, 2.63 ± 1.17%; 10 µM, 1.92 ± 0.81%; 100 µM, 1.26 ± 0.22%, Figure 4A,B). Furthermore, aloesone decreased the ratio of LPS-induced cells in the late phase of apoptosis from 16.66 ± 0.21% to 9.13 ± 1.38% (0.1 µM), 12.61 ± 1.99% (1 µM), 9.80 ± 2.35% (10 µM), and 7.89 ± 2.02% (100 µM, Figure 4A,C).

### 2.5. Mammalian Target of Rapamycin (mTOR)/Hypoxia Inducible Factor-1α (HIF-1α) and TLR4 Are Involved in the Protective Effects of Aloesone Post LPS Stimulation

To explain the molecular mechanism of aloesone in oxidative stress, inflammation, M1 polarization, and apoptosis, overlapping genes were collected from Genecards and SwissTargetsPrediction. The results demonstrated that 86 genes were closely associated with the antioxidant stress, anti-inflammation, anti-polarization, and anti-apoptotic effects of aloesone (Figure 5A,B), of which seven targets—heat shock protein HSP 90-alpha (HSP90AA1), signal transducer and activator of transcription 3 (Stat3), mitogen-activated protein kinase 1 (Mapk1), mTOR, fyn proto-oncogene (Fyn), protein tyrosine kinase 2 beta (Ptk2b), and lck proto-oncogene (Lck)—were the hub genes (Figure 5C). Moreover, pathway enrichment analysis demonstrated that these seven hub genes were enriched in Th17 cell differentiation, PD-L1 expression, and PD-1 checkpoint pathways in cancer, acute myeloid leukemia, pancreatic cancer, epidermal growth factor receptor (EGFR) tyrosine kinase inhibitor resistance, natural killer cell-mediated cytotoxicity, prostate cancer, T cell receptor signaling pathway, hypoxia inducible factor (HIF)-1 signaling, and phospholipase D signaling pathway (Figure 5D). 

Results from immunofluorescent staining (IF) illustrated that aloesone significantly repressed the LPS-induced activation of mTOR (Figure 6A,B), p-mTOR (Figure 6A,C), and HIF-1α (Figure 6A,D). Furthermore, aloesone decreased the membrane expression of TLR4, the specific receptor of LPS (Figure 6E). These results suggest that the mTOR/HIF-1α pathway and TLR4 may be involved in the protective effects of aloesone.

## 3. Discussion

Traditional medicine (TM) is commonly used worldwide. According to the prediction of the World Health Organization (WHO), 80% of the global population utilizes TM as a complementary or alternative medicine [32]. To date, many herbal extracts and specific natural compounds have shown anti-inflammation and antioxidant effects, including those in A. vera [33,34]. In the present study, we demonstrated that aloesone, a major metabolic compound of A. vera, has multiple protective effects against oxidative stress, inflammation, M1 polarization, and apoptosis. 

Macrophages play a vital role in the pathogenesis of many chronic diseases, including fibrosis, asthma, and inflammatory bowel disease [35]. Macrophages are an important source of many key inflammatory cytokines that drive autoimmune inflammation, such as IL-12, IL-18, IL-23, and TNF-α. ROS are normally produced within the body in limited quantities and are important compounds involved in the regulatory processes of cell homeostasis and functions, including signal transduction, gene expression, and receptor activation [36]. An imbalance in ROS results in oxidative stress, which induces inflammation by damaging DNA, proteins, and lipids [37,38]. Herein, synthesized aloesone repressed LPS-stimulated ROS production and induced the mRNA expression of vital antioxidant enzymes (Gpx1 and SOD1), suggesting the antioxidant stress effect of aloesone in RAW264.7 cells, which is consistent with a previous study in which aloesone scavenges radial DPPH and has high oxygen radical absorbance capacity at concentrations of 351 ± 35 μM and 66 ± 1 μM Trolox equivalents, respectively, in vitro [23]. 

Excessive ROS produced in the process of oxidant metabolism, as well as some natural or artificial chemicals, have been reported to stimulate macrophage M1 polarization, the proinflammatory phenotype [39], and subsequently initiate the inflammatory process. M1 macrophages tend to promote the synthesis and secretion of proinflammatory cytokines, such as iNOS, IL-1ꞵ, and TNF-α. These cytokines have also been documented to play critical roles in the inflammatory process, especially by causing macrophages apoptosis [40], leading to several chronic diseases [41]. In the present study, aloesone inhibited the M1 polarization of macrophages and alleviated the LPS-stimulated excessive release of NO and the overexpression of iNOS, IL-1ꞵ, and TNF-α in these cells, illustrating its anti-inflammatory effect. Furthermore, aloesone suppressed both the early and late phase of apoptosis. Overall, the multiple beneficial effects of aloesone on macrophages were confirmed. 

Elucidating the mechanism of aloesone is vital for its further application. The mammalian target rapamycin (mTOR) is a serine/threonine kinase involved in gene regulation in inflammation [42]. The phosphorylation of mTOR can regulate the phosphorylation of various transcription factors, including p70S6K and 4E-BP1, which can further promote the expression of HIF-1α [43]. The mTOR/HIF-1α pathway participates in cellular responses, such as survival and polarization [44,45,46]. Aloesone inhibits the mTOR/HIF-1α pathway, which could be one of the potential mechanisms involved in its therapeutic effects regarding inflammation, oxidative stress, polarization, and apoptosis in RAW264.7 cells. Moreover, LPS binds to TLR4, leading to oxidative stress, inflammation, and M1 polarization. In the present study, we demonstrated that aloesone significantly decreased the membrane expression of TLR4, which can also be regulated by HIF-1α [47]. In contrast, the facilitation of HIF-1α by LPS is regulated by TLR4 [48]. Moreover, LPS-induced oxidative stress could induce HIF-1α expression and is central to determining the phenotype of macrophages [49]. Therefore, HIF-1α may be the core mediator of aloesone in protecting macrophages from oxidative stress, inflammation, polarization, and apoptosis.

## 4. Materials and Methods

### 4.1. Synthesis of Aloesone

Aloesone was synthesized in accordance with the methods of a previous study [50], and was obtained as a white solid powder. In brief, β-diketone was derived from an acetophenone derivative, by coupling it with 1, 3-dioxolane-proted acetoacetic acid, followed by treating it with hydrochloric acid and isopropanol to afford the aloesone. Nuclear magnetic resonance (NMR) spectra were obtained on a 400 MHz ECZ400S spectrometer (JEOL, Tokyo, Japan, 400 MHz for ^1^H and 100 MHz for ^13^C): ^1^H NMR (400 MHz, dimethyl sulfoxide (DMSO)-d6) *δ* 10.61 (s, 1H), 6.62 (s, 1H), 6.60 (s, 1H), 6.03 (s, 1H), 3.85 (s, 2H), 2.65 (s, 3H), 2.21 (s, 3H); ^13^C NMR (100 MHz, DMSO-d6) *δ* 203.37, 178.69, 161.57, 161.10, 159.75, 142.13, 117.19, 114.88, 113.41, 101.06, 47.91, 30.38, 22.97 (Figure 7). Aloesone was dissolved in DMSO to prepare a stock solution (100 mM).

### 4.2. Study Design

#### 4.2.1. Cell Culture

RAW264.7 cells were purchased from Procell (Wuhan, Hubei, China) and cultivated in high glucose Dulbecco’s Modified Eagle Medium (DMEM; Gibco, Carlsbad, CA, USA), with the addition of 10% fetal bovine serum (CLARK Bioscience, Richmond, VA, USA) and 1% streptomycin/penicillin antibiotics (Biosharp, Hefei, China) in a humidified incubator with a stable CO_2_ (5%) supply and temperature (37 °C).

#### 4.2.2. Cell Viability

RAW264.7 cells (10,000 cells) were inoculated in 96-well microplates for 24 h and treated with 0.1, 1, 10, 100, and 1000 µM aloesone for 12 h. After removing the culture medium, cells were incubated with fresh culture medium containing 20 µL CCK8 (Biosharp) reagent for 30 min at 37 °C. Cell proliferation was evaluated by measuring the absorbance (450 nm) in a microplate reader (Spectra MAX 190, Molecular Devices, San Jose, CA, USA), according to the previously described method [51]. The survival rate was calculated by OD_aloesone_/OD_control_ × 100%.

#### 4.2.3. Groups

RAW264.7 cells were divided into six groups, as follows:(1)Control group: Cells were treated for 2 h with DMEM containing 0.1% DMSO as a vehicle, followed by DMEM with 0.1 M phosphate buffered saline (PBS, Gibco, New York, NY, USA) for 12 h.(2)LPS group: Cells were treated with DMEM containing 0.1% DMSO for 2 h, followed by DMEM with 1 µg/mL of LPS for an additional 12 h.(3)Aloesone groups: Cells were pretreated with various concentrations (0.1, 1, 10, and 100 µM) of aloesone in DMEM for 2 h, followed by DMEM with 1 µg/mL of LPS for an additional 12 h (Figure 8).

### 4.3. Evaluation of Oxidative Stress

#### 4.3.1. Measurement of ROS Generation 

ROS accumulation was detected using a radical probe, 2,7-dichlorodi-hydrofluorescein diacetate (DCFH-DA, Sigma, Saint Louis, MO, USA). RAW264.7 cells pretreated with aloesone and LPS were incubated with DCFH-DA (diluted to 1:1000 with serum-free medium) at 37 °C for 30 min in the dark. Then, the excess DCFH-DA that had not entered the cells was cleared using PBS. Thereafter, graphs were obtained by a magnification microscope (Zeiss X-Cite, Oberkochen, BW, Germany) and the mean fluorescence intensity was measured using flow cytometry (Agilent NovoCyte, Santa Clara, CA, USA) at the fluorescein isothiocyanate (FITC) channel.

#### 4.3.2. Quantitative Real-Time Polymerase Chain Reaction (qRT-PCR)

Total RNA was extracted from RAW264.7 cells using TRIzol Reagent (Invitrogen, Waltham, MA, USA), according to the manufacturer’s instructions. Then, it was reversed to cDNA in a total volume of 20 µL (HiScript Reverse Transcrptase, Vazyme, Nanjing, China). PCR was performed using a real-time PCR system (Bio-Rad, Hercules, CA, USA), with the following amplification conditions: 95 °C initial denaturation for 5 min, followed by 39 cycles of 95 °C for 15 s and 60 °C for 30 s. Relative expression levels of the antioxidant enzymes *Gpx-1* and *SOD-1* were calculated based on the 2^−∆∆Ct^ method, according to the manufacturer’s specifications, using the *actin* gene as a reference housekeeping gene. The sequences of primers used for qRT-PCR are shown below (Table 1), according to a previous study [52].

### 4.4. Assessment of Inflammation

#### 4.4.1. Detection of NO

Supernatant was used to detect the content of NO using a commercial kit based on the Griess reaction (Beyotime, Shanghai, China), as described previously [53]. The reaction was measured at 450 nm using a microplate reader (Spectra MAX 190).

#### 4.4.2. Detection of mRNA Expression of Inflammation Associated Genes 

Relative expression levels of inflammatory cytokines, including *iNOS*, *IL-1ꞵ*, and *TNF-α*, were detected by qRT-PCR and calculated based on the 2^−∆∆Ct^ method, according to the manufacturer’s specifications, using the *actin* gene as a reference housekeeping gene.

### 4.5. Evaluation of Macrophage Polarization

The cells were washed with 0.1 M PBS, harvested using trypsin (Biosharp, Hefei, China), and centrifuged at 1500 rpm for 5 min at 4 °C. Aliquots of 100,000 cells were suspended in PBS and incubated with an FITC-conjugated monoclonal antibody against M1 marker CD86 (1 μg, Abcam, Cambridge, UK). The cells were resuspended with PBS and analyzed using a flow cytometer. 

### 4.6. Detection of Apoptosis

The anti-appotic effect of aloesone was evaluated using the annexin V-FITC/propidium iodide (PI) apoptosis assay. Briefly, cells were harvested using trypsin and centrifuged at 1500 rpm for 5 min at 4 °C. Aliquots of 100,000 cells were suspended in 500 µL binding buffer and 5 µL staining reagent (Boster, Wuhan, Hubei, China). After incubation in the dark at 37 °C for 5 min, the fluorescent intensity of FITC and PI were analyzed by flow cytometry [54].

### 4.7. Predicting Targets and Pathways of Aloesone

The canonical simplified molecular input line entry system (SMILES) of aloesone were retrieved from PubChem (https://pubchem.ncbi.nlm.nih.gov/, accessed on 28 September 2022) and used for target identification in the Swiss Target Prediction database (http://www.swisstargetprediction.ch, accessed on 28 September 2022). Simultaneously, the target genes associated with inflammation, oxidative stress, macrophage polarization, and apoptosis were obtained from GeneCards (https://www.genecards.org/, version 4.9.0, accessed on 28 September 2022). Overlapping genes associated with aloesone, inflammation, oxidative stress, macrophage polarization, and apoptosis were retrieved using the EVenn (http://www.ehbio.com/test/venn/#/, accessed on 28 September 2022) tool [55], while the protein–protein interaction (PPI) network of the target was obtained using the STRING online tool (https://string-db.org/, accessed on 28 September 2022). Core genes were analyzed using CytoNCA of Cytoscape3.6.1 (https://www.cytoscape.org/, accessed on 28 September 2022), with the criteria of median value of degree centrality (DC), closeness centrality (CC), and betweenness centrality (BC) [56]. Biochemical pathways enriched by the core targets were determined using the web-based annotation tool DAVID v6.8 (https://david.ncifcrf.gov/tools.jsp, accessed on 28 September 2022). Statistical significance was set at *p* < 0.05.

### 4.8. Confirmation of Targets of Aloesone

#### 4.8.1. IF Staining

The expression of the targets (mTOR, p-mTOR, and HIF-1α) was detected by IF. Briefly, cells mounted on slides were retrieved and incubated for 30 min in 5% bovine serum albumin and incubated with rabbit monoclonal antibodies overnight at 4 °C (mTOR, 1:50, Abcam; p-mTOR, 1:50, Santa Cruz, CA, USA; and HIF-1α, 1:50, Abcam). Next, the cells were washed with PBS and stained with an FITC-conjugated secondary antibody (Boster) for 2 h at 37 °C, followed by 4′,6-diamidino-2-phenylindole (DAPI, Boster) for 5 min in the dark. A magnification microscope (Zeiss X-Cite) was used to observe the stained cells at a 200× magnification [24,57]. Subsequently, ImageJ software (1.52a, Wayne Rasband, Bethesda, MD, USA) was used to process the fluorescent images [58].

#### 4.8.2. Membrane Distribution of TLR4

Cells were washed with PBS. After the cells were collected, the phycoerythrin-conjugated monoclonal antibody of TLR4 (1 μg per 100,000 cells, Santa Cruz) were applied to stain the cells. The cells were resuspended with PBS and analyzed using a flow cytometer.

### 4.9. Statistical Analysis

All data are expressed as mean ± standard deviation (SD). The normal distribution of data was tested using the Shapiro–Wilk test, after which data that distributed normally were analyzed by one-way analysis of variance (ANOVA) with Benjamini’s test for multiple groups. Otherwise, non-normally distributed data were analyzed using the Kruskal–Wallis test. The results were considered to be significant with *p* < 0.05. Statistical analysis and figures were generated using GraphPad Prism (version 9.0.0).

## 5. Conclusions

In summary, we first demonstrated that aloesone significantly inhibited ROS production, NO release, and surface expression of CD86, and suppressed the early and late phase of apoptosis in RAW264.7 cells; this confirmed the multiple protective effects of aloesone on oxidative stress, inflammation, M1 polarization, and apoptosis of the macrophages. Furthermore, the mTOR/HIF-1α pathway and TLR4 were closely related to these effects. This study confirmed the potential use of aloesone as a therapeutic agent.

## Figures and Tables

**Figure 1 molecules-28-01617-f001:**
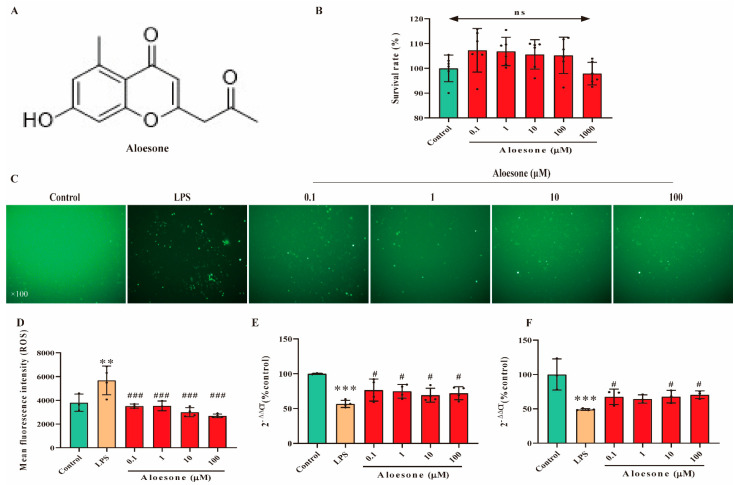
Antioxidant effect of aloesone in RAW264.7. Cells were treated with aloesone for 2 h, followed by 1 µg/mL of LPS for an additional 12 h. (**A**) Chemical structure of aloesone. (**B**) The effect of aloesone administered at concentrations ranging from 0.1 to 1000 µM for 2 h on the survival rate of RAW264.7 cells, *n* = 6. (**C**) Fluorescent figures of 2, 7-dichlorodi-hydrofluorescein (DCFH) among treatment groups. (**D**) Mean fluorescent intensity analyzed by flow cytometry, *n* = 3–4. (**E**) The mRNA expression of antioxidant enzyme Gpx-1. (**F**) The mRNA expression of antioxidant enzyme SOD-1.; ^#^
*p* < 0.05 compared with the LPS group; *n* = 3–4. Results were expressed as mean ± standard deviation, ** *p* < 0.01, *** *p* < 0.001 compared with the control group; ^#^ *p* < 0.05, ^###^ *p* < 0.001 compared with the LPS group. ns: no significance.

**Figure 2 molecules-28-01617-f002:**
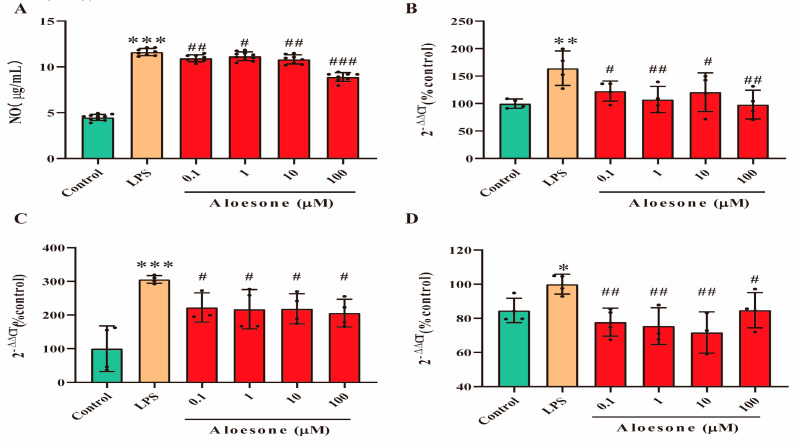
Anti-inflammatory effects of aloesone on LPS-induced RAW264.7 cells. Cells were treated with aloesone for 2 h, followed by 1 µg/mL of LPS for an additional 12 h. (**A**) Release of NO, *n* = 6. (**B**) mRNA expression of inflammatory cytokine *iNOS*, *n* = 3–4. (**C**) mRNA expression of *IL-1ꞵ*, *n* = 3–4 (**D**) mRNA expression of *TNF-α*, *n* = 3–4. Results were expressed as mean ± standard deviation, * *p* < 0.05, ** *p* < 0.01, *** *p* < 0.001 compared with the control group; ^#^ *p* < 0.05, ^##^ *p* < 0.01, ^###^ *p* < 0.001 compared with the LPS group.

**Figure 3 molecules-28-01617-f003:**
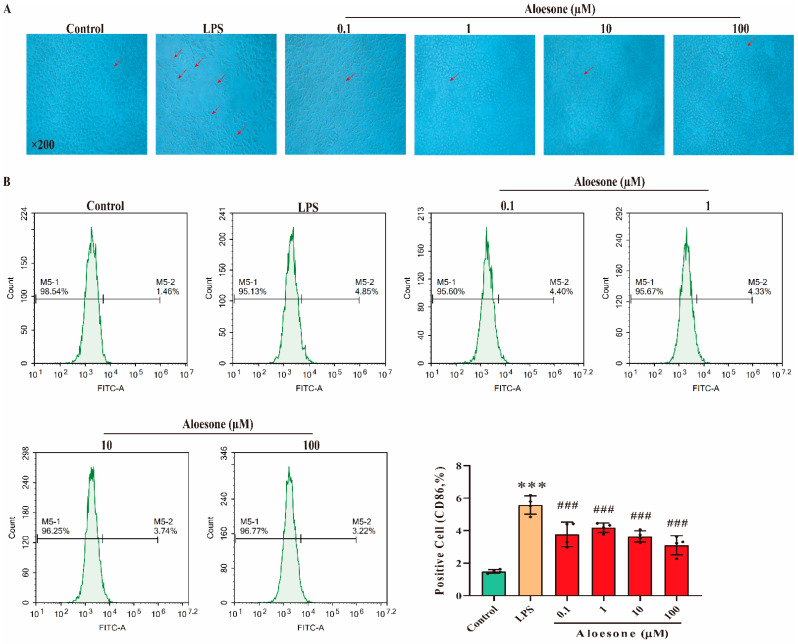
Aloesone inhibited the polarization of RAW264.7 to M1 macrophages when stimulated by LPS, *n* = 4. (**A**) The representative graphs of RAW264.7 when administered with LPS alone or with aloesone. (**B**) Percentage of CD86-positive cells among treatment groups. Results were expressed as mean ± standard deviation, *** *p* < 0.001 compared with the control group; ^###^ *p* < 0.001 compared with the LPS group.

**Figure 4 molecules-28-01617-f004:**
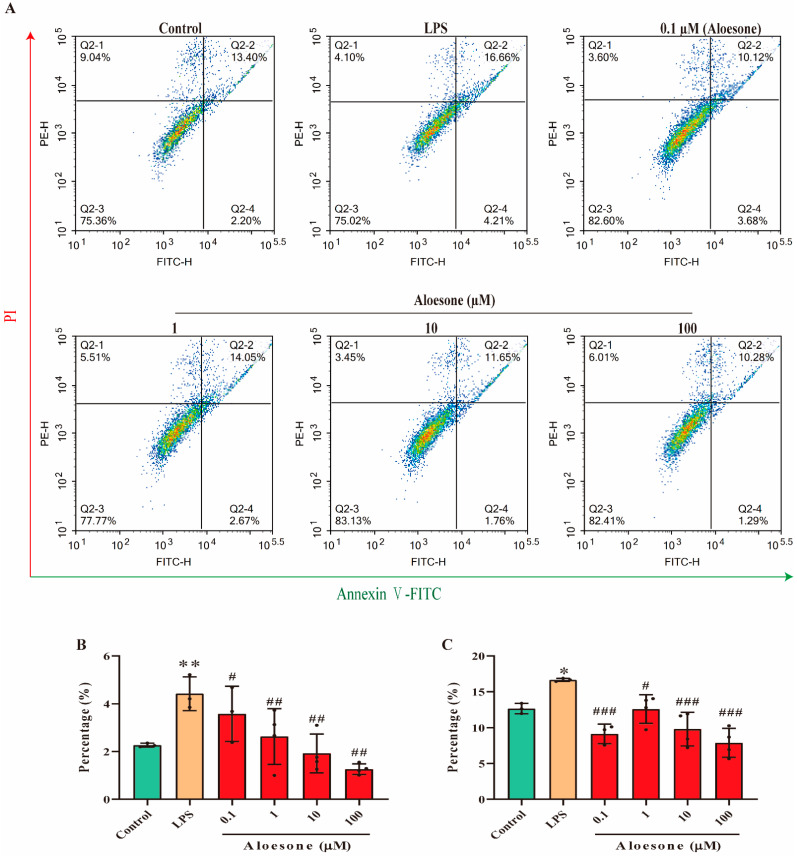
Aloesone inhibited the LPS-induced early and late apoptosis of RAW264.7 cells, *n* = 3–4. Cells were treated with aloesone for 2 h, followed by 1 µg/mL of LPS for an additional 12 h. (**A**) Representative graphs from flow cytometry. (**B**) Early phase apoptotic rate. (**C**) Late phase apoptotic rate. Results were expressed as mean ± standard deviation, * *p* < 0.05, ** *p* < 0.01 compared with the control group; ^#^ *p* < 0.05, ^##^ *p* < 0.01, ^###^ *p* < 0.001 compared with the LPS group.

**Figure 5 molecules-28-01617-f005:**
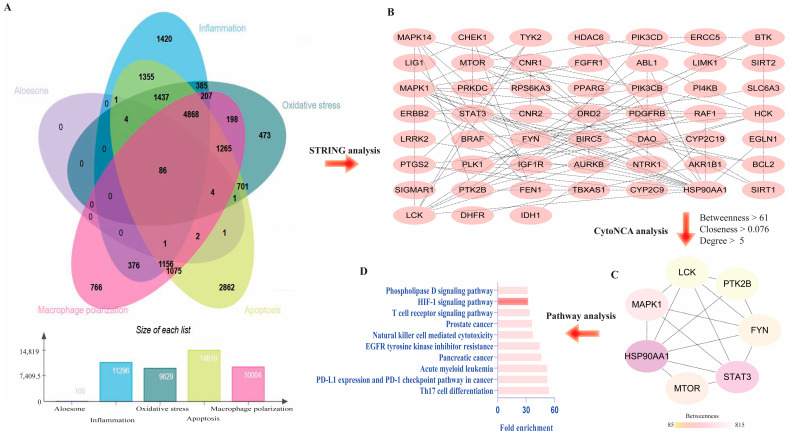
Predicted targets of aloesone. (**A**) Venn diagram of genes affected by aloesone and those involved in oxidative stress, inflammation, M1 polarization, and apoptosis. (**B**) Overlapping genes (86 genes). (**C**) Hub genes. Core genes were analyzed using CytoNCA of Cytoscape3.6.1, with the criteria of median value of betweenness, closeness, and degree centrality. (**D**) Enrichment pathway of hub genes.

**Figure 6 molecules-28-01617-f006:**
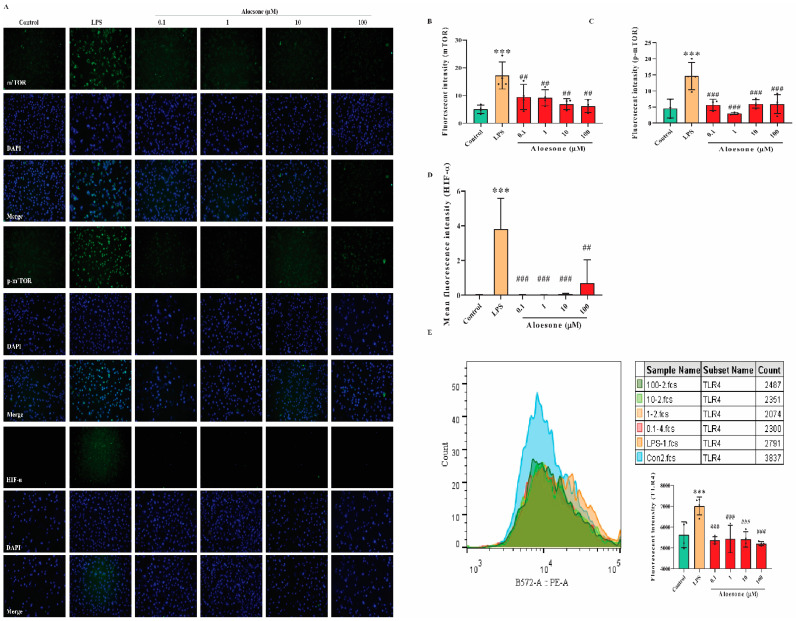
Potential molecular mechanism of aloesone. (**A**) Immunofluorescent figures of mTOR, p-mTOR, and HIF-1α expression. Cells were treated with aloesone for 2 h, followed by 1 µg/mL of LPS for an additional 12 h. (**B**–**D**) Fluorescent intensity of mTOR, p-mTOR, and HIF-1α, respectively; *n* = 3–4. (**E**) Fluorescent intensity of TLR4 in the control, LPS, 0.1, 1, 10, and 100 µM group; *n* = 3–4. Results were expressed as mean ± standard deviation, *** *p* < 0.001 compared with the control group; ^##^ *p* < 0.01, ^###^ *p* < 0.001 compared with the LPS group.

**Figure 7 molecules-28-01617-f007:**
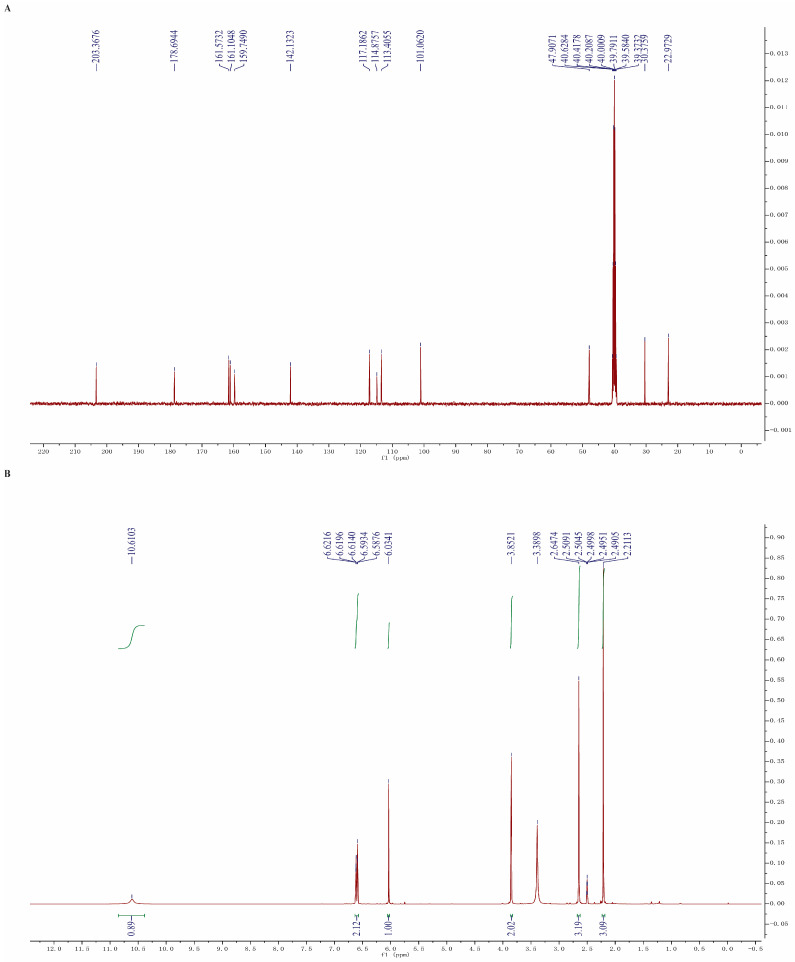
NMR spectra of aloesone. (**A**) ^13^C NMR. (**B**) ^1^H NMR.

**Figure 8 molecules-28-01617-f008:**
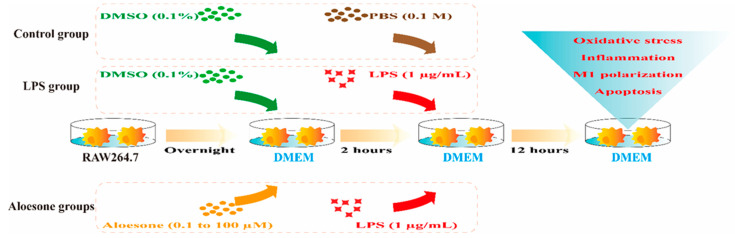
Administration of aloesone to RAW264.7 cells.

**Table 1 molecules-28-01617-t001:** Sequence of primers.

Gene Names	Primer-F (5′-3′)	Primer-R (5′-3′)	Length (bp)
*Actin*	CCACAGCTGAGAGGGAAATC	AAGGAAGGCTGGAAAAGAGC	193
*iNOS*	TTGGGTCTTGTTCACTCCACG	GGCTGAGAACAGCACAAGGG	201
*IL-1ꞵ*	TGCCACCTTTTGACAGTGATG	GGAGCCTGTAGTGCAGTTGT	351
*TNF-α*	GTAGCCCACGTCGTAGCAA	GTGAGGAGCACGTAGTCGG	191
*Gpx-1*	GGAGAATGGCAAGAATGAAGA	CCGCAGGAAGGTAAAGAG	139
*SOD-1*	CCATCAGTATGGGGACAATACA	GGTCTCCAACATGCCTCTCT	109

## Data Availability

The data presented in this study are available on request from the corresponding author.

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
