# Peer review of "Multiple Beneficial Effects of Aloesone from Aloe vera on LPS-Induced RAW264.7 Cells, Including the Inhibition of Oxidative Stress, Inflammation, M1 Polarization, and Apoptosis"

_molecules, 2023, doi:10.3390/molecules28041617_

Round 1

Reviewer 1 Report

Introduction: do not use a free line to separate parts of introduction,

I miss some information about the cell line.

Line 46: which studies? You mentioned one study.

Explain abbreviations when noticed for the first time (check whole manuscript)

If It is not necessary, do not use italics in the results section – see instructions for authors.

References – check instructions for authors and rewrite. I recommend using Template.

Lines 79-88: this is very confusing. I don’t think that synthesis of aloesone was mentioned as the aim of this study, so why it is in the results section? Is this part of another study?

Lines 79-82: this should be moved to material and methods. Describe the principle and briefly the procedure of the aloesone synthesis.

Lines 89-90: this is also methodology, not result.

Figure 2: fix name of Y axis in 2A; 2B – this is methodology.

Fix Figure 2B so that the font is not so wide and put it to „study design“. Use „lock aspect ratio“ when editing graphs – whole manuscript

Do not discuss in the results section..lines 91-95

Do not use hyphen in „anti-oxidative“

There are asterisks, hash marks, but I dont really understand what mean the black dots in bar graphs.. ?

I recommend using the names of sub-sections like „the release of NO and mRNA expression..do not tell reader the whole message in the title of subsection.

Lines 110-111, 123-128, 161-162: Don’t discuss in the results section. Just describe the results.

If it is possible, make the scales in Figure 2B uniform, so the difference between group would be more visible.

Line 149, 176: use „antioxidant, anti-inflammatory, anti-apoptotic...“ check whole manuscript – including abstract and keywords.

Discussion:

Notice the Figures in the text within the results section, not in the discussion.

Line 190: „...previous studies.“ Which?? Cite them.

Line 191: His surname is not Francisco. And they did not mention aloesone in their publication. This is misinterpretation. Fix it.

Line 192: Did you read the cited publication? The presence of aloe vera did not influence mRNA levels for catalase when compared to fresh or cryopreserved control.

Line 194: radical DPPH, oxygen-radical absorbing capacity

Line 195: μM trolox equivalents is the name of unit.

Methodology: I recommend to make separate section concerning study design. Add full names of chemicals when noticed for the first time.  

Line 242-243: this is part of study design, not the methodology of nitric oxide assay

Line 250: correct aloesone

Add information: PBS – manufacturer, volume, centrifugation – add RCF

Please, be more precise in interpreting your results in the conclusion section. And also when concluding within the abstract.

How many repetitions were performed? Why did you choose 1 μg/mL of LPS?

Last but not least, review by English native speaker. 

Author Response

We thank the reviewer for the detailed and valuable comments, which have helped us improve this study. We have revised the manuscript accordingly with care to correct the errors that you have mentioned. We hope that the revised document meets your approval and will be suitable for publication.

  1. Do not use a free line to separate parts of introduction

Response: We apologize for these errors. In this revised manuscript, we have deleted the free lines.

  1. I miss some information about the cell line.

Response: The murine macrophage cell line RAW264.7 was used in the present study. This information has also been added in the revised manuscript (Lines 77-78), as following:

Macrophages are a central component of the innate immune system in peripheral and play a vital role in inflammation [1]. The membranes of Gram-negative bacteria contain LPS, which activates the host Toll-like receptor 4 (TLR4) receptor and triggers an inflammatory response, ultimately, leading to the release of pro-inflammatory mediators. The LPS-induced murine macrophage cell line RAW264.7 model has been commonly used to explore inflammation, oxidative stress, and apoptosis [2, 3].

  1. Line 46: which studies? You mentioned one study. Line 190: „...previous studies.“ Which?? Cite them.

Response: Thank you for your oberservation. We have amended line 46 to reflect the one referenced study and revised the corresponding sentence of line 190,  as following:

Line 46: Recent studies have confirmed that ROS could also cause oxidative stress, leading to the activation of inflammatory pathways, stimulation of macrophage polarization, and triggering cell damage [4, 5].

Line 190: Herein, synthesized aloesone repressed LPS-stimulated ROS production and induced the mRNA expression of vital antioxidant enzymes (Gpx1 and SOD1), suggesting the antioxidant stress effect of aloesone in RAW264.7 cells, which is consistent with a previous study in which aloesone scavenges radial DPPH and has high oxygen radical absorbance capacity at concentrations of 351 ± 35 μM and 66 ± 1 μM Trolox equivalents, respectively, in vitro [6].

  1. Explain abbreviations when noticed for the first time (check whole manuscript)

Response: We have revised all abbreviations at first mention in the revised manuscript.

  1. If It is not necessary, do not use italics in the results section – see instructions for authors

Response: Thank you for the comment. We have revised the results section to normal font.

  1. References – check instructions for authors and rewrite. I recommend using Template.

Response: We have edited the references according to the Molecules template.

  1. Lines 79-88: this is very confusing. I don’t think that synthesis of aloesone was mentioned as the aim of this study, so why it is in the results section? Is this part of another study? Lines 79-82: this should be moved to material and methods. Describe the principle and briefly the procedure of the aloesone synthesis

Response: Thank you for your helpful comments. The synthesis of aloesone is the base of present study. To make the reader understand the complete study details, we decided to add the synthesis section in the manuscript. Further, we have added the following in the discussion to highlight the synthesis of aloesone “Herein, synthesized aloesone repressed LPS-stimulated ROS production and induced the mRNA expression of vital antioxidant enzymes (Gpx1 and SOD1), suggesting the antioxidant stress effect of aloesone in RAW264.7 cells, which is consistent with previous studies.”

As per your comments, we have also added a brief synthesis process of aloesone and moved the content and figure on the synthesis of aloesone to the material and methods section, as following:

Aloesone was synthesized in accordance with the methods of a previous study [7], and obtained as a white solid powder. In brief, β-diketone was derived from an acetophenone derivative, coupling with 1, 3-dioxolane-proted acetoacetic acid, followed by treated with hydrochloric acid and isopropanol to afford the aloesone.

  1. Lines 89-90: this is also methodology, not result.

Response: Line 89-90 describes the mRNA expression of Gpx-1 and SOD-1 which was detected by realtime PCR and how it is closely associated with oxidant stress.

  1. Figure 2: fix name of Y axis in 2A; 2B – this is methodology. Fix Figure 2B so that the font is not so wide and put it to „study design“. Use „lock aspect ratio“ when editing graphs – whole manuscript. If it is possible, make the scales in Figure 2B uniform, so the difference between group would be more visible.

Response: The label on the Y axis in Figure 2A has been replaced with survival rate (%), while Figure 2B has been moved to the material and methods section.

  1. Do not discuss in the results section. lines 91-95. Lines 110-111, 123-128, 161-162: Don’t discuss in the results section. Just describe the results.

Response: Thank you for your comments. We have deleted the corresponding sentences in the revised manuscript.

  1. Do not use hyphen in „anti-oxidative“. Line 149, 176: use „antioxidant, anti-inflammatory, anti-apoptotic...“ check whole manuscript–including abstract and keywords.

Response: We have revised for consistency throughout the revised manuscript.

  1. There are asterisks, hash marks, but I don’t really understand what mean the black dots in bar graphs.. ? How many repetitions were performed? Why did you choose 1 μg/mL of LPS?

Response: Bar-plots were used to show the numerical values, while the black dots represent each value of detection thus showing the repetitions perfomed in the study. Moreover, the repetitions in each subsection have been added to the figure legends.

The LPS-induced murine macrophage cell line RAW264.7 model has been commonly used to explore inflammation, oxidative stress, and apoptosis [2]. Moreover, the concentration of 1 μg/mL of LPS was shown to significantly induce inflammation, oxidative, and polarization in previous studies [8-10]. Hence, in this study, we applied 1 μg/mL of LPS to explore the effects of aloesone.

  1. I recommend using the names of sub-sections like „the release of NO and mRNA expression. Do not tell reader the whole message in the title of subsection.

Response: We have revised the names of sub-sections, as following:

2.1. Aloesone inhibited oxidative stress induced by LPS

2.2. Aloesone suppressed inflammation induced by LPS in RAW264.7 cells

2.3. Aloesone inhibited the M1-polarization of RAW 264.7 cells induced by LPS

2.4. Aloesone suppressed LPS-induced apoptosis in RAW 264.7 cells

2.5. Mammalian target of rapamycin (mTOR)/hypoxia inducible factor-1α (HIF-1α) and TLR4 involved in the protective effects of aloesone post LPS stimulation

  1. Notice the Figures in the text within the results section, not in the discussion.

Response: We have moved the graphic abstract below the conclusion.

  1. Line 191: His surname is not Francisco. And they did not mention aloesone in their publication. This is misinterpretation. Fix it. Line 192: Did you read the cited publication? The presence of aloe vera did not influence mRNA levels for catalase when compared to fresh or cryopreserved control.

Response: We apologize for the misunderstanding reference and very thanks for your comments. As the reference focused on 10% extract of A. Vera but not for aloesone exactly, we have moved the sentence to the line 172-173 to clearly read as follows: To date, many herbal extracts and specific natural compounds have shown anti-inflammation and antioxidant effects, including those in A. vera [11, 12].

  1. Line 194: radical DPPH, oxygen-radical absorbing capacity. Line 195: μM trolox equivalents is the name of unit

Response: We have added trolox equivalents as the units accordingly.

  1. Methodology: I recommend to make separate section concerning study design. Add full names of chemicals when noticed for the first time.

Response: Thank you for your comments. We have rearranged the material and methods section into nine subsections, including synthesis of aloesone, study design, evaluation of oxidative stress, assessment of inflammation, evaluation of macrophage polarization, detection of apoptosis, predicting targets and pathways of aloesone, confirmation of targets of aloesone, and statistical analysis. Full names of chemicals also have been shown when noticed for the first time.

  1. Line 242-243: this is part of study design, not the methodology of nitric oxide assay

Response: We have revised the methodology of nitric oxide assay accordingly in the revised manuscript.

  1. Line 250: correct aloesone

Response: We have revised the sentence in line 300-304 of the revised manuscript.

  1. Add information: PBS – manufacturer, volume, centrifugation – add RCF

Response: We have added the recommended information on PBS (line 250) and RCF of centrifugation (line 268 and 292) in the revised manuscript.

  1. Please, be more precise in interpreting your results in the conclusion section. And also when concluding within the abstract.

Response: Thank you for your comments. We have re-checked and revised the elucidation of the results in the revised manuscript.

  1. Last but not least, review by English native speaker.

Response: The revised manuscript had been reviewed by an English language native speaker from Editage.

Reviewer 2 Report

The manuscript titled “Multiple beneficial effects of aloesone from Aloe vera on LPS-induced RAW264.7 cells: inhibits oxidative stress, inflammation, M1 polarization, and apoptosis ”  show the influence of aloesone on inhibiting oxidative stress, inflammation, M1 polarization, and apoptosis. The manuscript was well-written, and appropriate methods were used and good results were obtained in this study.

However, the authors should provide more detailed description for the purpose of this study (Lines 73-77), the acquisition and identification of aloesone (Lines 81-85). Moreover, the conclusion seems too simple.  

Minor comments

Line 15: [email protected],; One more comma

Line 52: Aloe vera, when the name of a species appears for the second time, it should be abbreviated as A. vera. After that, the species names in other place should be changed to A. vera.

Line 58: According to the rules of MDPI, numbers 0-9 should be written as words unless they are a measurement. So 8 in the Line should be eight. Please check and correct similar problems in the full text.

Line 86: Figure 1 (B and C), Please provide clearer pictures (300dpi).

Lines 229, 236, 267, 268, 301: There is a writing error in this temperature unit.

Line 293: P < 0.05 should changed to P < 0.05.

References 26, 32, 34, 36, 37: These references lack page numbers or article numbers.

References 8, 11, 15, 16, 17, 23, 35: The species or genus name (Aloe vera) needs to be italicized.

In addition, Please confirm whether the first letter of each word in the title of the reference article should be capitalized. If not, please be consistent.

Author Response

We thank the reviewer for the more detailed and valuable comments. We have revised the manuscript accordingly with care to correct the errors that you have mentioned. We hope that the revised document meets your approval and is suitable for publication.

Main points:

The authors should provide more detailed description for the purpose of this study (Lines 73-77), the acquisition and identification of aloesone (Lines 81-85). Moreover, the conclusion seems too simple.  

Response: Detailed information on the purpose of study had been added in the introduction section. Further, based on your comments, a brief description of the acquisition and identification of aloesone have been included in the material and methods section. The conclusion has also been enriched as follows: 

Purpose of this study: The aim of the present study was to explore the effects of aloesone on LPS-induced macrophages by evaluating the main markers of oxidative stress, inflammation, polarization, and apoptosis. The molecular mechanism of aloesone was further studied.

Synthesis of aloesone: Aloesone was synthesized in accordance with the methods of a previous study [7], and obtained as a white solid powder. In brief, β-diketone was derived from an acetophenone derivative, coupling with 1, 3-dioxolane-proted acetoacetic acid, followed by treated with hydrochloric acid and isopropanol to afford the aloesone. Neclear magnetic resonance (NMR) spectra were obtained on a 400 MHz ECZ400S spectrometer (JEOL, Tokyo, Japan, 400 MHz for 1H and 100 MHz for 13C). 1H NMR (400 MHz, dimethyl sulfoxide (DMSO)-d6) ? 10.61 (s, 1H), 6.62 (s, 1H), 6.60 (s, 1H), 6.03 (s, 1H), 3.85 (s, 2H), 2.65 (s, 3H), 2.21 (s, 3H). 13C NMR (100 MHz, DMSO-d6) ? 203.37, 178.69, 161.57, 161.10, 159.75, 142.13, 117.19, 114.88, 113.41, 101.06, 47.91, 30.38, 22.97 (Figure 7). Aloesone was dissolved in 0.1% of DMSO to prepare a stock solution (100 mM).

Conclusion: In summary, we firstly demonstrated that aloesone significantly inhibited ROS production, NO release, surface expression of CD86, and suppressed the early and late phase of apoptosis in RAW264.7 cell, which confirmed the multiple protective effects of aloesone on oxidative stress, inflammation, M1 polarization, and apoptosis of macrophages. Furthermore, the mTOR/HIF-1α pathway and TLR4 were closely related to these effects. This study confirmed the potential use of aloesone as a therapeutic agent.

Minor points:

  1. [email protected],; One more comma

Response: We have revised this error in the manuscript.

  1. Aloe vera, when the name of a species appears for the second time, it should be abbreviated as A. vera. After that, the species names in other place should be changed to A. vera..

Response: We thank you for your comments. We have abbreviated the species name following the first mention in the entire manuscript.

  1. According to the rules of MDPI, numbers 0-9 should be written as words unless they are a measurement. So 8 in the Line should be eight. Please check and correct similar problems in the full text.

Response: We have revised it and checked for consistency in the revised manuscript.

  1. Line 86: Figure 1 (B and C), Please provide clearer pictures (300dpi). Lines 229, 236, 267, 268, 301: There is a writing error in this temperature unit.

Response: We have revised figure 1 and checked these errors for consistency in the revised manuscript.

  1. Line 293: P < 0.05 should changed to P < 0.05.

Response: We have revised it and checked for consistency in the revised manuscript.

  1. References 26, 32, 34, 36, 37: These references lack page numbers or article numbers.References 8, 11, 15, 16, 17, 23, 35: The species or genus name (Aloe vera) needs to be italicized.In addition, Please confirm whether the first letter of each word in the title of the reference article should be capitalized. If not, please be consistent.

Response: We have revised all references for accuarcy and consistency.  

Round 2

Reviewer 1 Report

Line 222: ..followed by treating

Line223: Nuclear magnetic..

Line228: You meant..aloesone was dissolved in DMSO, while the final concentration of DMSO in each group was 0.1% ...? Fix it, please. 

Line247: Cells were pretreated...

Line249: ..were defined

Line249: DMSO was replaced with aloesone before LPS was administered in the LPS group? Please, explain this sentence to me and try to fix it in the manuscript.

Lines 323-326: add information about concentrations of monoclonal antibodies and cells. 

Author Response

We thank the reviewer for the detailed and valuable comments due to our unsatisfactory responses in the previous round. We have revised the manuscript accordingly with care to correct the errors that you have mentioned. We hope that the revised document meets your approval and is suitable for publication.

  1. Line 222: ..followed by treating. Line223: Nuclear magnetic.. Line249: ..were defined

Response: We apologize for these errors. We have made the necessary corrections in the revised manuscript.

  1. Line228: You meant..aloesone was dissolved in DMSO, while the final concentration of DMSO in each group was 0.1% ...? Fix it, please. 

Response: Thank you for your valuable comments and we apologize for the error in expression. The stock solution of aloesone (100 mM) was dissolved in DMSO, then diluted to 0.1, 1, 10, and 100 μM with DMEM (containing 10% FBS and 1% streptomycin/penicillin antibiotics. As the highest concentration of DMSO was 0.1% in our study, we applied 0.1% DMSO as the vehicle control to eliminate the influence of solvent (DMSO). We have revised the text in line 228, as follows:

Aloesone was dissolved in DMSO to prepare a stock solution (100 mM)

  1. Line249: DMSO was replaced with aloesone before LPS was administered in the LPS group? Please, explain this sentence to me and try to fix it in the manuscript.

Response: Thank you for your comments. We have modified the description of the groups in the revised manuscript (lines 248-257) to improve readability, as follows:

The RAW264.7 cells were divided into six groups, as follows:

Control group: Cells were treated for 2 h with DMEM containing 0.1% DMSO as a vehicle, followed by DMEM with 0.1 M phosphate buffered saline (PBS, Gibco, New York, US) for 12 h.

LPS group: Cells were treated with DMEM containing 0.1% DMSO for 2 h, followed by DMEM with 1 µg/mL of LPS for an additional 12 h.

Aloesone groups: Cells were pretreated with various concentrations (0.1, 1, 10, and 100 µM) of aloesone in DMEM for 2 h, followed by DMEM with 1 µg/mL of LPS for an additional 12 h (Figure 8).